# Carbon opportunity cost increases carbon footprint advantage of grain-finished beef

**Daniel Blaustein-Rejto**[ID][1]*, **Nicole Soltis**[2], **Linus Blomqvist**[1,3]

**1** The Breakthrough Institute, Berkeley, California, United States of America, **2** California Council on Science and Technology, Sacramento, California, United States of America, **3** Bren School of Environmental Science & Management, University of California, Santa Barbara, California, United States of America

* dan@thebreakthrough.org

**Data Availability Statement:** All relevant data are within the paper and its Supporting information files.

## Abstract

Beef production accounts for the largest share of global livestock greenhouse gas emissions and is an important target for climate mitigation efforts. Most life-cycle assessments comparing the carbon footprint of beef production systems have been limited to production emissions. None also consider potential carbon sequestration due to grazing and alternate uses of land used for production. We assess the carbon footprint of 100 beef production systems in 16 countries, including production emissions, soil carbon sequestration from grazing, and carbon opportunity cost—the potential carbon sequestration that could occur on land if it were not used for production. We conduct a pairwise comparison of pasture-finished operations in which cattle almost exclusively consume grasses and forage, and grain-finished operations in which cattle are first grazed and then fed a grain-based diet. We find that pasture-finished operations have 20% higher production emissions and 42% higher carbon footprint than grain-finished systems. We also find that more land-intensive operations generally have higher carbon footprints. Regression analysis indicates that a 10% increase in land-use intensity is associated with a 4.8% increase in production emissions, but a 9.0% increase in carbon footprint, including production emissions, soil carbon sequestration and carbon opportunity cost. The carbon opportunity cost of operations was, on average, 130% larger than production emissions. These results point to the importance of accounting for carbon opportunity cost in assessing the sustainability of beef production systems and developing climate mitigation strategies.

## Introduction

Beef production accounts for about 6% of all anthropogenic greenhouse gas emissions [1]. Given rising demand in developing countries, reducing the greenhouse-gas (or carbon) footprint of production, measured as kilograms carbon dioxide-equivalent ($CO_2$e) per kilogram of beef, is an important climate mitigation strategy [2, 3].

Whether beef is produced in pasture-finished or grain-finished systems affects its carbon footprint. In both pasture-finished and grain-finished systems, cattle are raised initially on pasture or rangeland. The primary difference lies in the finishing stage—in grain-finished systems, cattle are fed a grain-based diet and often kept in feedlots, whereas cattle in pasture-

**Funding:** The authors received no specific funding for this work.

**Competing interests:** The authors have declared that no competing interests exist.

finished systems continue to eat fresh and stored grasses and hay until they reach slaughter weight [4]. The finishing stage therefore accounts for any potential difference in the carbon footprint of these systems. Pasture-finished systems are common in many parts of the world and account for approximately 33% of global beef production. Grain-finished systems account for 15%, and other systems, such as mixed crop-livestock production, account for the remainder [5].

Most life-cycle assessments of the carbon footprint of grain-finished and pasture-finished systems have been limited to emissions directly attributable to cradle-to-farmgate activities (here referred to as production emissions) [6]. Reviews and meta-analyses of these studies conclude that pasture-finished systems have higher average production emissions [4, 6, 7]. Grain finishing typically leads to much higher growth rates. As a result, proportionally less energy is expended on maintenance rather than growth, such that inputs and emissions per unit of beef is lower [8].

In addition to emissions associated with production, beef's carbon footprint is also influenced by land use. Recent meta-analyses show that pasture-finished systems have higher land-use intensity (measured as area per unit production) on average, since the amount of pasture needed in the finishing stage of pasture-finished cattle is much larger than the amount of cropland needed to provide grain for the finishing stage of grain-finished cattle [4, 6].

Greater land requirements influence the carbon footprint in two ways. First, pasture and crop management can increase soil carbon sequestration [9, 10]. Use of improved grazing practices in some pasture-finished systems has sequestered enough carbon to offset production emissions from finishing [11]. Yet large soil carbon sequestration rates are only possible under particular agro-ecological conditions and for a limited time period [9, 12].

Second, greater land use for beef production can displace native ecosystems and reduce land available for restoration. The amount of $CO_2$ that could be removed on land used for production through reforestation or other restoration has been referred to as the "carbon opportunity cost" [13].

Existing global comparisons of pasture-finished and grain-finished systems are incomplete as they do not account for both carbon opportunity cost and soil carbon sequestration. For instance, Poore and Nemecek (2018) [6], in a global meta-analysis of life-cycle assessments, do not account for potential soil carbon sequestration from production or the carbon opportunity cost of land use. The authors do account for emissions from land-use change, but only from recent changes in which total area for the crop or livestock product increased in the country of production. This approach, unlike the carbon opportunity cost approach, can result in zero carbon costs associated with many types of land use (see Searchinger *et al.* 2018 [14] Supplementary Discussion for a detailed treatment). Balmford *et al.* (2018) [15] estimate the relationship between the carbon footprint and land-use intensity of beef production including foregone carbon sequestration from land use—finding that there is a strong positive correlation—but their analysis is limited to Latin America and does not estimate soil carbon sequestration from grazing. Schmidinger and Stehfest (2012) [16], Searchinger *et al.* (2018) [14], and Hayek *et al.* (2020) [13] estimate the carbon opportunity cost of beef production at different geographic scales, but do not compare grain-finished and pasture-finished systems or estimate soil carbon sequestration from grazing.

Here, for the first time, we assess the sum of production emissions, soil carbon sequestration, and carbon opportunity cost–referred to here as the carbon footprint–of pasture-finished and grain-finished systems from across the world. We compare the carbon footprint of pasture-finished and grain-finished systems that exist in the same region and that have been studied using the same methodology. We also use regression analysis to assess the relationship between land-use intensity and carbon footprint, regardless of the system.

Beef production systems are changing rapidly across the world, and decisions about the future direction of this change will have important implications for climate mitigation as well as other environmental impacts. Accounting for the carbon footprint, including the carbon opportunity cost, as we do in this paper, should help guide these decisions.

## Materials and methods

We calculate the carbon footprint (the sum of production emissions, soil carbon sequestration, and carbon opportunity costs in kilograms $CO_2$e per kilogram of retail weight beef) of 100 beef production operations across 16 countries, including those from beef and dairy herds, drawn from a dataset of food and beverage life-cycle assessments [6] and from Stanley *et al.* (2018) [11]. Poore and Nemecek (2018) [6] includes production emissions and land-use intensity data. Stanley *et al.* (2018) [11] reports production emissions, carbon sequestration, emissions from soil erosion, and land-use intensity for the finishing stage of a pasture-finished and grain-finished operation in the Midwestern USA; we derive values from earlier stages from Pelletier *et al.* (2010) [17] which also studied operations in the Midwest. We conduct a pairwise comparison of carbon footprints between pasture-finished and grain-finished beef production systems, and a regression analysis of the relationship between land-use intensity and carbon footprint.

### Production emissions and land-use intensity

Production emissions represent cradle-to-farmgate life-cycle greenhouse gas emissions. This includes emissions associated with enteric fermentation, animal housing, manure management, and inputs associated with feed production such as fertilizers, pesticides, and machinery.

Land-use intensity represents land required for grazing and crop production, in hectare per kilogram of retail weight beef. Land use for pasture is calculated as the sum of temporary and permanent pasture, and land use for cropland is calculated as the sum of seed, arable and fallowed crop land. We use and standardize production emissions and land-use intensity values from Poore and Nemecek (2018) [6] and Stanley *et al.* (2018) [11].

### Soil carbon sequestration

Soil carbon sequestration (SCS) in kg $CO_2$ per kg of retail weight beef is calculated as the product of land-use intensity of grazing (LUI) and carbon sequestration due to grazing (CSG) in kg C ha$^{-1}$ yr$^{-1}$ (Eq 1).

$$SCS = LUI \cdot CS \cdot \frac{44 \, CO_2}{12 \, C} \tag{1}$$

There is insufficient data to calculate a specific carbon sequestration rate for each life-cycle assessment location. This is in part because sequestration rates depend on environmental and management factors, such as soil texture and grazing intensity, not consistently described in the life-cycle assessments. Instead, for all life-cycle assessments we use the mean carbon sequestration rate of 0.28 Mg C ha$^{-1}$ yr$^{-1}$ for "improved grazing management" estimated in a synthesis of the grassland management literature [18]. This estimate, drawn from studies with an average soil depth of 23 cm, is within the range of peer reviewed estimates: 0.03 and 1.04 Mg C ha$^{-1}$yr$^{-1}$, with the lowest values corresponding to dry climates and the highest to specific grassland management practices and regions [19]. Our use of a single mean rate for diverse locations could lead to us overestimating the relationship between land use intensity and carbon footprint if actual sequestration rates on grazed land in the studies we include are greater

than 0.28 Mg C ha$^{-1}$ yr$^{-1}$. However, given that not all the life-cycle assessments included are of operations with improved grazing practices, the true carbon sequestration rates across operations may be lower. To be conservative in our carbon footprint for grain-finished operations, we assume that no carbon sequestration occurs on cropland used for feed production, consistent with research that shows that $CO_2$ emissions from agricultural land are generally balanced by removals [20].

## Carbon opportunity cost

Our measure of carbon opportunity cost calculates how much carbon sequestration would have occurred had land been occupied with native ecosystems instead of pasture or cropland. This assumes that reducing land-use intensity results in proportionately less agricultural land area locally.

We calculate carbon opportunity cost (COC) as the sum of the carbon opportunity cost of pasture ($p$) and cropland ($c$) used in production. For each of these two land uses, the carbon opportunity cost is calculated as the product of land-use intensity (LUI) and potential carbon sequestration (PCS) of the land in the area where the life-cycle assessments was conducted, in kg C ha$^{-1}$ yr$^{-1}$ (Eqs 2 and 3).

$$COC = \sum_i LUI_i \cdot PCS_i \cdot \frac{44\,CO_2}{12\,C} \text{ for } i = c, p \tag{2}$$

where

$$PCS_i = \frac{NPP_i \cdot k_i \cdot r - s_i}{r} \text{ for } i = c, p \tag{3}$$

$NPP_i$ denotes the potential net primary productivity of native vegetation (kg C ha$^{-1}$ yr$^{-1}$) that could be restored on agricultural land within a given radius of where the life-cycle assessment was conducted. We report results using a radius of 2 degrees (~223 km at equator). $k_i$ is the conversion factor from net primary productivity to carbon sequestration in vegetation and soils or, put differently, the average level of carbon sequestration generated by devoting one kilogram of NPP to restoring native vegetation. This value is 0.42 kg $CO_2$ ha$^{-1}$ yr$^{-1}$ for every kg of NPP for cropland and 0.44 for pasture, as calculated by Searchinger *et al.* (2018) [14]. $r$ denotes the time period over which carbon sequestration is averaged, in this case 100 years; and $s_i$ denotes existing vegetation carbon stocks (kg C ha$^{-1}$), 1100 for cropland and 3100 for pasture, based on global averages for cereals and pasture, respectively, from Searchinger *et al.* (2018) [14]. Although spatially explicit estimates of cropland carbon stocks exist [21], we are not aware of any for pasture carbon stocks.

The logic behind Eq 3 is as follows. The numerator represents the difference in potential carbon stocks between current land use and native vegetation. $NPP_i \cdot k_i$ is a flux measure, in kilograms of carbon per hectare per year, which we multiply by 100 to turn into a stock measure. In effect, this assumes that the equilibrium carbon stock in native ecosystem is reached after 100 years. The numerator, the difference in potential carbon stocks, is then divided by 100 to arrive at an annual (flux) rate. We select a time period of 100 years because this is roughly the age at which forest stands can be considered mature and the carbon stock becomes relatively stable, and the time period used in Searchinger *et al.* (2018) [14] and Schmidinger and Stehfest (2012) [16] to calculate average carbon sequestration rates in regenerating forests.

Data on potential net primary productivity under native vegetation is generated by the Lund–Potsdam–Jena managed Land (LPJmL) model, a dynamic global vegetation model that

simulates vegetation composition, distribution, and carbon stocks and flows at 0.5x0.5˚ spatial resolution. We use LPJmL results from Searchinger *et al.* (2018) [14].

We assume life-cycle assessment sites located in climate categorized as "dry" in Poore & Nemecek (2018) [6] have zero potential carbon sequestration because they either cannot support substantial additional biomass or are native grasslands or savannas for which restoration does not typically involve reforestation [22].

## Pairwise comparison between pasture-finished and grain-finished production systems

We compare the carbon footprint of 20 pairs of pasture-finished and grain-finished production systems, across 12 countries, in the Poore and Nemecek (2018) [6] database and one recent comparative life-cycle assessment [11] with and without soil carbon sequestration and carbon opportunity cost included. Systems were selected for inclusion if they were in the same subnational region or country, if the study was national in scope, and reported in the same study or within two studies by the same primary author. Details of the pairs are listed in S8 Table in S1 File. Fourteen of the pairs were reported for the same geographic region, but lacked coordinates. For those, we estimated carbon opportunity cost by calculating mean potential net primary productivity on cropland and grazing land within the subnational region or country the life-cycle assessment was located (Supplementary Methods in S1 File). We used a paired t-test to test if the mean difference between the pasture-finished and grain-finished system was significantly different from zero.

## Regression analysis

We also assess the relationship between carbon footprint and land-use intensity using cross-section regression analysis of beef production operations. We include 72 operations from life-cycle assessments that report geographic coordinates, including a total of 24 studies in 12 countries (S1 Fig and S7 Table in S1 File). We log-transform the carbon footprint and land-use intensity because the input data is heavily right-skewed and because this enables us to present results as elasticities—the expected percent change in the carbon footprint with a percentage change in land-use intensity.

We run three different regressions, starting with production emissions as the only regressor, adding carbon opportunity cost in the second regression, and then also including soil carbon sequestration in the third regression. We use a linear model to facilitate comparison of the relationship across the regressions. Since there may be variables operating at the country level that influence the carbon footprint (e.g. climate, national policy), we use a multilevel model with country-level random effects, particularly varying intercepts and constant slopes [23]. This yields the following regression equation:

$$log\left(carbon\,footprint_{i,j}\right) = \beta_0 + \beta_1 log\left(LUI_{i,j}\right) + u_j + \epsilon_{i,j} \qquad (4)$$

where j indexes countries, i indexes operations within countries, $\beta_0 + u_j$ is the intercept for each country, $\beta_1$ represents the elasticity between land-use intensity and the carbon footprint, and $\epsilon_{ij}$ is an error term.

We choose this specification over a fixed effect model as there is substantial variation in the independent variable within units (i.e. countries), the level of correlation between unit effects and the independent variable is not extremely high, and we are interested in accounting for the variability between units but not in estimating specific unit effects, in which case a random effects model can be appropriate to use and result in superior estimates [24]. Regressions with

fixed effects produced results very similar to those with random effects (S5 Table in S1 File). Our analysis examines differences in carbon footprints across operations with different land-use intensity and does not attempt causal inference per se.

## Robustness checks

We vary four parameters to assess the robustness of the results. First, we run the analysis with 0.25, 0.5, 1.0 and 4.0 degree radius. We do this to confirm our results cannot be explained by the choice of radius as NPP values can vary widely over a small area.

Second, we run the analysis with alternative calculations for carbon opportunity cost at the national and global levels. The national and global carbon opportunity costs assume that if the amount of land needed to support a given level of food production declines by one unit as a result of lower land-use intensity, then one unit of land will be restored to native vegetation somewhere in the country or world, respectively. These are relevant comparisons in cases where domestic and international trade allow land-use intensity reductions to be spatially disconnected from pasture and cropland expansion/contraction. We calculate national carbon opportunity cost using the average NPP values over all crop and pasture land across the country each production system is located in. This method could be improved by using crop-specific values; however, not all life-cycle assessments in our dataset describe which crops are used in production. We also calculate global carbon opportunity cost using average global net primary production values.

Third, we run the analysis using a carbon sequestration rate of 0.47 Mg C ha$^{-1}$ yr$^{-1}$, the average value reported across all studies of improved grassland management included in Conant et al. (2017) [18]. This reduces the carbon footprint of more land-intensive operations such as pasture-finished systems more than it reduces the carbon footprint of less land-intensive operations.

Fourth, we run the analysis with and without the potential carbon sequestration, and thus the carbon opportunity cost, set to 0 for operations in dry climates.

## Results

In this study we calculated the carbon footprint of beef production systems as the sum of production emissions, carbon opportunity cost, and soil carbon sequestration, and assessed the relationship of this carbon footprint measure and land-use intensity. After presenting summary statistics, we show the results of the pair-wise comparison of the carbon footprints of pasture-finished and grain-finished beef production systems. We then present results from regression analysis of different measures of carbon footprints, with and without carbon opportunity cost and soil carbon sequestration, on land-use intensity.

The carbon footprint, including production emissions, carbon opportunity cost, and soil carbon sequestration, across the 72 beef production operations with reported latitude and longitude, and the 28 operations without latitude/longitude included in the pasture-finished/grain-finished comparison ranged from -68.3 to 2169.3 kg $CO_2$e kg$^{-1}$ retail weight, with mean 177.37 and median 107.14 (Table 1). The wide range is due to the diversity in environmental and management conditions. The two operations with the largest carbon footprint values are pasture-finished with degraded or nominal pasture and low or no pasture management, and among the highest land use intensity values. Four pasture-finished and one grain-finished production systems in Queensland, Australia are estimated to have negative carbon footprints, in part because we assume that the dry climate results in zero carbon opportunity cost. If soil carbon sequestration rates are lower in dry climates than other climates, as some studies such as Smith et al. (2008) [20] suggest, these operations would be more likely to also have positive

**Table 1. Summary statistics for beef operations.**

| Variable | Mean | Median | Range | SD | CV | 95% CI | Units |
|---|---|---|---|---|---|---|---|
| Production emissions | 52.64 | 41.42 | 4.9, 182 | 36.1 | 0.69 | 45.48, 59.8 | kg $CO_2$e kg$^{-1}$ |
| Soil carbon sequestration | -15.11 | -7.41 | -164.8, 0 | 24.4 | -1.62 | -19.96, -10.26 | kg $CO_2$e kg$^{-1}$ |
| Carbon opportunity cost | 139.85 | 68.46 | 0, 2243 | 266.0 | 1.9 | 87.1, 192.59 | kg $CO_2$e kg$^{-1}$ |
| Carbon footprint | 177.37 | 107.14 | -68.3, 2169.3 | 26.0 | 1.49 | 124.79, 229.96 | kg $CO_2$e kg$^{-1}$ |
| Land-use intensity | 0.02 | 0.01 | 0, 0.2 | 0.02 | 1.27 | 0.01, 0.02 | ha kg$^{-1}$ |

All units are per kilogram retail weight. $n = 100$.

carbon footprints. The carbon footprint was similar in robustness checks, with the mean value ranging from 141.6 to 210.0 kg $CO_2$e kg$^{-1}$ retail weight when different radii are used and when we do not assume zero carbon opportunity cost for arid climates (S1 Table in S1 File).

In individual systems, carbon opportunity cost was, on average, 130% larger than production emissions. Soil carbon sequestration offset 31.5% of production emissions and 18.9% of the production emissions and carbon opportunity cost, on average. Across all robustness checks, carbon opportunity cost is at least 65% larger than production emissions and soil carbon sequestration does not fully offset production emissions (S2 Table in S1 File).

## Pairwise comparison between pasture-finished and grain-finished systems

The pairwise comparison found that pasture-finished systems had 20% higher mean production emissions than grain-finished systems on average (p<0.01). When also including soil carbon sequestration, the difference is not statistically significant at a 95% confidence level (p≥0.05). When the carbon opportunity cost is also accounted for, however, the carbon footprint of pasture-finished systems is on average 42% higher than that of grain-finished systems (p<0.01) (Fig 1). Compared to grain-finished systems, pasture-finished systems also had 15% higher median production emissions (p<0.01) and carbon footprints (p<0.05), indicating that while the magnitude of the difference is sensitive to extreme values, the general finding of higher emissions is robust (S3 Table in S1 File).

The carbon footprint of pasture-finished systems, including production emissions, soil carbon sequestration and carbon opportunity cost, is higher than that of the grain-finished systems (p<0.05) in the majority of robustness tests (S4 Table in S1 File). Differences are not significant (p≥0.05) in some cases when a smaller radius or higher rate of soil carbon sequestration is used.

## Regression analysis

In the regression analysis, when only production emissions are regressed on land-use intensity, the coefficient is 0.48 (Fig 2A, Table 2). This can be interpreted as a 10% increase in land-use intensity being associated with a 4.8% increase in emissions. Less land-intensive systems typically have lower production emissions. Fig 2A shows the regression line with this slope, with the level adjusted by country. When adding in soil carbon sequestration, the coefficient is reduced to 0.32, indicating that soil carbon sequestration offsets a part of the production emissions (Table 2).

However, the relationship between carbon footprint, including carbon opportunity cost, and land-use intensity is stronger, with a coefficient of 0.90 (Table 2, Fig 2B). Hence, a 10% increase in land-use intensity is associated with a 9.0% increase in the carbon footprint of beef production. This near-proportional relationship is in part due to the large share of the carbon

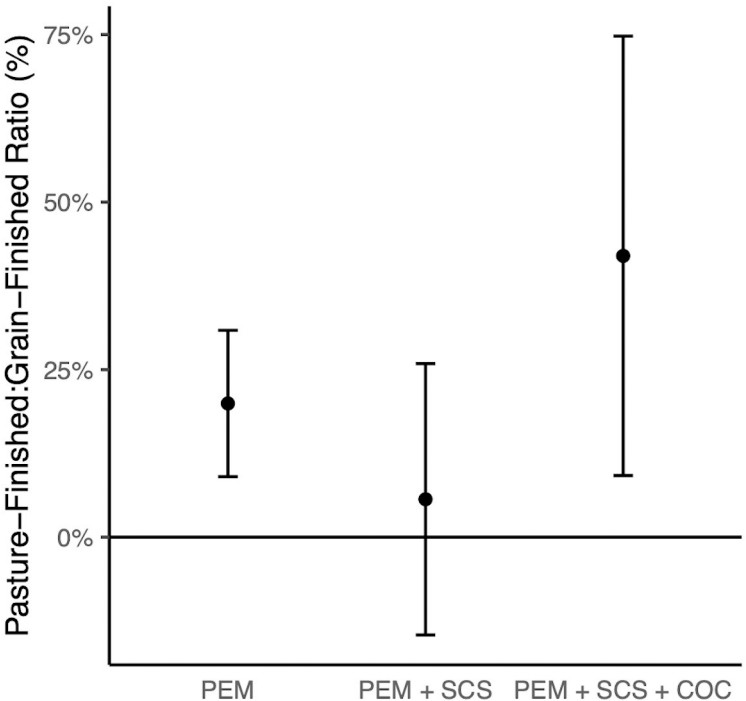

**Fig 1. Average ratios of carbon footprints between pasture-finished and grain-finished.** Ratios expressed as percentage difference. PEM denotes production emissions, SCS denotes soil carbon sequestration, and COC denotes carbon opportunity cost. Values above (below) 0 denote the carbon footprint for pasture-finished operations is larger (smaller) than for grain-finished operations. Comparisons were made within paired production systems to control for agronomic and environmental differences. Bars show means and 95% confidence intervals. On average, carbon footprints for pasture-finished operations are significantly greater (p<0.01) than those of grain-finished operations when only production emissions are included and when production emissions, soil carbon sequestration and carbon opportunity cost are included. *n* = 20 pairs.

footprint accounted for by carbon opportunity cost, which is proportional to land area in production.

Regressions with pooled and country fixed-effects specifications generate similar results (S5 Table in S1 File). Results are robust to other specifications and assumptions checked (S6 Table in S1 File).

## Discussion

Our analysis is the first global comparison of the carbon footprint of grain-finished and pasture-finished beef production systems that includes production emissions as well as soil carbon sequestration and carbon opportunity cost. This yields significant new insights that can inform environmental and agricultural decision-making.

Our results indicate that pasture-finished and other more land-intensive beef production systems have greater production emissions than grain-finished and less land-intensive systems. When we calculate carbon footprints including production emissions, soil carbon sequestration, and carbon opportunity cost, all beef production systems have a higher carbon footprint than when only production emissions are included, but pasture-finished systems have a substantially larger carbon footprint than grain-finished systems, and there is a strong positive relationship between land use intensity and carbon footprint.

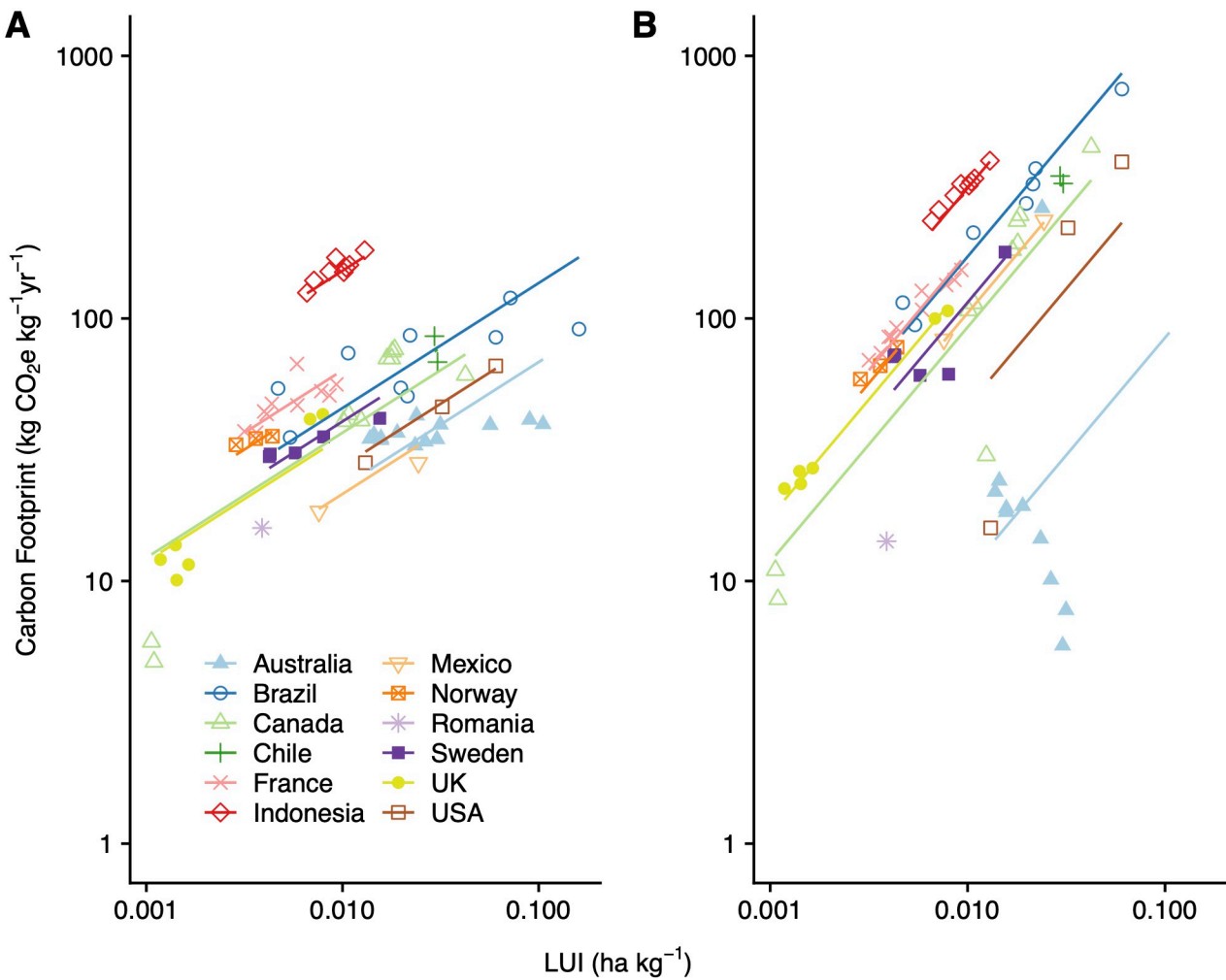

**Fig 2. The relationship between land-use intensity and carbon footprint of beef production systems.** Results from a regression of log(carbon footprint) on log(land-use intensity) with country random effects. Dots indicate life-cycle assessment observations; colors indicate countries; and lines represent the slope of the regression that includes all countries, adjusted according to the levels of each country. A) Carbon footprint including only production emissions. $n = 72$. B) Carbon footprint including production emissions, soil carbon sequestration and carbon opportunity cost. $n = 69$.

The differences in carbon footprint between pasture- and grain-finished operations are largely due to differences in carbon opportunity cost, which account for a large share of the total carbon footprint. The carbon opportunity cost of operations was, on average, 130% larger than production emissions. These results point to the importance of accounting for carbon opportunity cost in assessing the sustainability of beef production systems.

Our analysis also confirms that beef operations that have been studied in life-cycle assessments are generally not carbon neutral or negative. The mean carbon footprint across all studies, including production emissions, sequestration, and carbon opportunity cost, is over three times larger than the mean value for production emissions (Table 1). One exception is that we estimate negative carbon footprints for four grass-finished operations and one grain-finished operation that are in dry eco-climate zones in Australia, for which we assume there is zero carbon opportunity cost. This suggests that grazing cattle on dry rangeland with little to no carbon opportunity cost could have a small carbon footprint when the grazing also increases soil

**Table 2. Results from log-log regressions.**

| | Dependent variable: | | |
| --- | --- | --- | --- |
| | **PEM** | **PEM+SCS** | **PEM+SCS+COC** |
| LUI | 0.48*** | 0.32*** | 0.90*** |
| | (0.04) | (0.08) | (0.09) |
| Constant | 5.90*** | 4.84*** | 8.70*** |
| | (0.27) | (0.45) | (0.52) |
| Observations | 72 | 68 | 69 |
| $R^2$ | 0.67 | 0.27 | 0.63 |
| Adjusted $R^2$ | 0.66 | 0.25 | 0.63 |

Standard errors in parentheses. LUI = land-use intensity. PEM = production emissions. SCS = soil carbon sequestration. COC = carbon opportunity cost.

* $p < 0.1$,

** $p < 0.05$,

*** $p < 0.01$

organic carbon, as has been observed in some studies of dry rangeland with finer textured soil [12].

Our comparison of pasture-finished and grain-finished systems builds upon and strengthens past findings. Our finding that production emissions are 20% higher on pasture-finished operations than on grain-finished operations is consistent with Clark and Tilman (2017) [4], which found average emissions were 19% higher though their estimate was not statistically significant. In our results, soil carbon sequestration from grazing offsets only a portion of production emissions. This finding is consistent with the conclusions of Garnett *et al.* (2017) [19], which estimated that soil carbon sequestration from grazing can offset 20–60% of annual emissions from ruminant grazing.

Our finding that land-use intensity and carbon footprint are positively correlated strengthens similar findings from previous studies, none of which included production emissions, soil carbon sequestration and carbon opportunity cost, which is a more comprehensive approach for assessing the carbon footprint of land use than conventional land-use change approaches [14]. Poore and Nemecek (2018) [6] found that beef and lamb systems with lower land-use intensity have a lower carbon footprint when considering emissions from land-use change, but not carbon opportunity cost. Balmford *et al.* (2018) [15] used generalized linear mixed models to analyze the relationship between land-use intensity and carbon footprint, including a measure of carbon opportunity cost based on IPCC (2006) methods. Their analysis, limited to Brazil and tropical Mexico, also found that the carbon opportunity cost of agriculture was typically greater than production emissions, and that incorporating opportunity costs generated strongly positive associations between carbon footprint and land-use intensity. Searchinger *et al.* (2018) [14] calculated global-average carbon opportunity costs for beef similar to the average calculated for all operations included in this study. Their estimates of 165.3 and 143.9 kg $CO_2$e kg$^{-1}$ carcass weight were based on the potential carbon that could be gained or lost, respectively, on land used for production. The authors applied the values to five production systems in Brazil and found, consistent with our results, that systems with the lowest land-use intensity had the greatest carbon benefits.

Our study has several limitations although we do not believe these substantially alter our conclusions. The pairwise comparison of grain-finished and pasture-finished operations has a

relatively small sample of 20 pairs. This means that assumptions of asymptotic normality, which are the basis for the paired t-test, may not hold. However, our robustness checks (S4 Table in S1 File) and nonparametric test of the median (S3 Table in S1 File), which is robust to small sample sizes, extreme outliers, and heavy-tailed distributions, reinforce the conclusion that pasture-finished operations have greater production emissions and carbon footprints than grain-finished operations. In addition, our results cannot be considered to be globally representative or representative of all operations. The life-cycle assessments that underlie our study were not conducted to be globally representative. For instance, we include one study from Asia (Indonesia) and none from Africa. Nevertheless, given the consistent positive relationship between land use intensity and carbon footprint across operations in multiple geographies, we expect a similar relationship would be observed in other regions except in dry eco-climate zones where grazing can have little carbon opportunity cost.

In our study, we also assume that a change in land-use intensity results in a proportionate change in land under production and thus the land area with native ecosystems. While this has the advantage of simplicity, it is unlikely to be exactly true in reality, as a result of economic mechanisms. The real effect may be more or less than proportional depending, in part, on how differences in land-use intensity and carbon footprint are associated with total factor productivity. For instance, an operation shifting from grain-finished to pasture-finished may lower total factor productivity. This would increase prices and lead to a reduction in overall demand, while at the same time making that operation less profitable and thus induce producers elsewhere to produce more. The reduction in demand would reduce land use and the spillover of production would increase land use, with an ambiguous net impact.

It is also challenging to predict where a change in farmland area and native vegetation will take place as a result of changes in land-use intensity and production system in a given location. We calculate three measures of carbon opportunity cost: local, national, and global. These roughly correspond to different levels of market connectedness, which will differ between locations. For example, changes in US production can have large effects on global markets, whereas changes in less globally connected regions such as sub-Saharan Africa will likely see mostly local or national effects [25]. Furthermore, for those producers connected to global markets, effects of changes in production are not likely to be evenly distributed across the world, but are likely to be concentrated in those regions that are more globally integrated [25]. In the last few decades, much of the expansion of pasture has taken place in tropical countries like Brazil [26]. Following this logic, it is possible that higher land-use intensity in the US as a result of shifting to pasture-finished systems would displace production to these places, and is thus more likely to displace highly carbon-rich tropical ecosystems.

In addition, we use several simplifying assumptions. We use global mean estimates of soil carbon sequestration and current carbon stocks in cropland and grazing land vegetation due to lack of spatially-explicit data with global coverage. Our assumed rate is drawn from estimates for improved grazing management, so as to lessen the risk of overestimating the carbon footprint of grass-finished systems. Our measures of carbon opportunity cost are also based on mean potential carbon sequestration values in grazing land and cropland, if restored to native vegetation. They do not account for livestock diet rations, which crops are used for feed, or crop yields for instance. This may contribute to us underestimating potential carbon sequestration and carbon opportunity costs if feed crops such as soy are grown in areas with higher potential carbon sequestration, such as former forest, than other crops.

Future research could build upon our analysis by integrating more spatially explicit estimates of soil carbon sequestration and carbon stocks and calculating carbon opportunity cost based on how different cropland and grazing land is used in beef production. It could also incorporate additional types of environmental impacts and resource use, such as water use or

eutrophication potential, which are important in assessing the overall sustainability of production systems. Future research could also analyze the relationship between land use intensity and different greenhouse gases and incorporate different approaches to calculating their warming (e.g. GWP100, GWP20, GWP*) since each has a different atmospheric lifetime and effect on warming. Further types of beef and other livestock operations, such as pork or milk, could also be studied with similar methods.

Overall, this study provides a novel assessment of the carbon footprint of beef operations, building upon life-cycle assessments of production emissions to also include carbon sequestration and carbon opportunity cost. Our conclusion that beef operations with low land-use intensity, including grain-finished operations, have lower carbon footprints than pasture-finished operations and others with high land-use intensity provides important insights for agricultural stakeholders globally such as in Brazil where pasture expansion is a leading driver of forest loss [27]. Accounting for products' carbon opportunity cost, not just production emissions or soil carbon sequestration, could shift which production systems government programs, corporate procurement, investors, and consumers incentivize.

## Supporting information

**S1 File. Supplementary methods, figures and tables.**
(DOCX)

## Acknowledgments

The authors would like to thank Kenton de Kirby, Ken Cassman, and Joseph Poore for valuable comments on the draft manuscript.

## Author Contributions

**Conceptualization:** Daniel Blaustein-Rejto, Linus Blomqvist.

**Data curation:** Nicole Soltis.

**Formal analysis:** Daniel Blaustein-Rejto, Nicole Soltis.

**Methodology:** Daniel Blaustein-Rejto, Linus Blomqvist.

**Supervision:** Linus Blomqvist.

**Visualization:** Daniel Blaustein-Rejto.

**Writing – original draft:** Daniel Blaustein-Rejto.

**Writing – review & editing:** Daniel Blaustein-Rejto, Linus Blomqvist.

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
