## [Decision Letter · Decision Letter 0]

14 Sep 2022

PONE-D-22-23019Carbon opportunity cost increases carbon footprint advantage of grain-finished beefPLOS ONE

Dear Dr. Blaustein-Rejto,

Thank you for submitting your manuscript to PLOS ONE. After careful consideration, we feel that it has merit but does not fully meet PLOS ONE’s publication criteria as it currently stands. Therefore, we invite you to submit a revised version of the manuscript that addresses the points raised during the review process.

We look forward to receiving your revised manuscript.

Kind regards,

Juan Carlos Suárez Salazar

Academic Editor

PLOS ONE

2. We note that Figure S1 in your submission contain [map/satellite] images which may be copyrighted. All PLOS content is published under the Creative Commons Attribution License (CC BY 4.0), which means that the manuscript, images, and Supporting Information files will be freely available online, and any third party is permitted to access, download, copy, distribute, and use these materials in any way, even commercially, with proper attribution. For these reasons, we cannot publish previously copyrighted maps or satellite images created using proprietary data, such as Google software (Google Maps, Street View, and Earth). For more information, see our copyright guidelines: http://journals.plos.org/plosone/s/licenses-and-copyright.

A. You may seek permission from the original copyright holder of Figure S1 to publish the content specifically under the CC BY 4.0 license. 

B. If you are unable to obtain permission from the original copyright holder to publish these figures under the CC BY 4.0 license or if the copyright holder’s requirements are incompatible with the CC BY 4.0 license, please either i) remove the figure or ii) supply a replacement figure that complies with the CC BY 4.0 license. Please check copyright information on all replacement figures and update the figure caption with source information. If applicable, please specify in the figure caption text when a figure is similar but not identical to the original image and is therefore for illustrative purposes only.

Additional Editor Comments :

In general, the manuscript presents an adequate structure with some typographical errors. It is necessary that you can add additional information and explain more extensively some techniques of statistical analysis which has been requested by the reviewers, this will strengthen the manuscript.

Reviewers' comments:

Reviewer's Responses to Questions

**Comments to the Author**

1. Is the manuscript technically sound, and do the data support the conclusions?

Reviewer #1: Yes

Reviewer #2: Partly

Reviewer #3: Yes

2. Has the statistical analysis been performed appropriately and rigorously? 

Reviewer #1: Yes

Reviewer #2: No

Reviewer #3: Yes

3. Have the authors made all data underlying the findings in their manuscript fully available?

Reviewer #1: Yes

Reviewer #2: No

Reviewer #3: Yes

4. Is the manuscript presented in an intelligible fashion and written in standard English?

Reviewer #1: Yes

Reviewer #2: Yes

Reviewer #3: Yes

5. Review Comments to the Author

Reviewer #1: 1. Check the title.

40-43. Cite this part.

47-49. Cite this part.

89-94. These results should not be there.

130, 154, 157. Is it possible to reference the equations?

281 & 312. Please place figure title below the Fig.

320. Please include all the definitions below the table. The table must be self explanatory.

Figure S1. There are no white circles, and the map has no color

Recommendation: you may consider placing different types of dots on figure 2. This can help colorblind (daltonic) people with their understanding.

Reviewer #2: The paper conducts a pairwise comparison of pasture-finished operations and grain-finished operations. Also, the authors show that land-use intensity and carbon footprint are positively correlated. Thus, the paper concerns a fundamental and interesting research problem.

However, I have some recommendations for the authors:

1. Materials and methods.

a. Pairwise comparison: How did you collect the data? Table S8 shows information about 20 studies included in the pairwise comparison; how were the studies selected? It is necessary to add this information.

b. Regression analysis: How many observations did you use? Are the units 72 operations or 24 studies? Is it a cross-section regression, or is it panel data? (In this case, which are the individuals, and which is the time?) Where are the results of the Hausman test? Is it possible to use the Hausman test in a cross-section model? It is important to explain which model was estimated and what type of data was used.

2. Results and discussion

a. Pairwise comparison: This section describes the main findings from the t-test, but it seems that the numbers in table S3 are different from the text. Why didn’t you compare the findings from the nonparametric test with the t-test? If your data was collected from studies, why didn’t you use meta-analysis inside the t-test? I suggest explaining how the studies were selected and considering making a meta-analysis.

b. Regression analysis. Were the assumptions of the model validated? Are the same results if the authors add other variables (control)? Are the number of observations enough to estimate this kind of model?

Reviewer #3: This paper presents a rather complete comparison of the carbon footprint of grain-finished and pasture-finished beef production Systems, taking into account production emissions, soil carbon sequestration and carbon opportunity cost. The study use global information from 16 countries and several model to estimate emissions are considered. It is generally well written, with very few typing errors. The statistical methods used are correct and the authors took into account assumptions that make them valid. I recommend publishing it once a few minor corrections have been done.

Minor corrections:

Line 36. Replace “ kg” by “ kilograms”

Lines 76-76 and 80. Replace “et al” by “et al.”

Line 107 . Replace “ kg” by “ kilograms”

Line 109 111 113 Replace “et al” by “et al.”

Line 117 Replace “&” by “and”

Line 121 Replace “ kg” by “ kilograms” and eliminate “(ha)”. Universal acronyms like “ha” should not be defined

Line 125 Replace “et al” by “et al.”

Line 135 Replace “0.28 Mg carbon (C) ha-1 yr-1 “ by “0.28 Mg C ha-1 yr-1 “

Line 156 Replace “Where” by “where”

Line 162 Replace “one kg” by “one kilogram” or by “1 kg”

Lines 164 and 167 Replace “et al” by “et al.”

Line 172 Replace “ kg” by “ kilograms”

In lines 70, 76, 80 and 81 among others you write the author’s name, year, and the number of citation between brackets, but in several lines like 176, 189, 239, 259 (and more) the number in brackets is absent.

Line 176 Replace “et al” by “et al.”

Line 206 “where” without capital letter

Line 237 Replace “MgC” by “Mg C”

Line 239 Replace “et al” by “et al.”

Line 254 and 261. Replace “kg CO2e/kg” by “kg CO2e kg-1”

Line 259 Replace “et al” by “et al.”

Line 264 Table 1. Eliminate the dot at the end of table title

Line 265 Replace “ kg” by “ kilograms”

Line 320 Table 2. explain what the numbers in parentheses are

Line 353 Replace “et al” by “et al.”

Line 362 Replace “et al” by “et al.”

Line 482-483 Replace Agricultural Systems by the abbreviation Agric. Syst.

Table S4: Replace “kgCO2e kg-1” by “kg CO2e kg-1”

6. PLOS authors have the option to publish the peer review history of their article (what does this mean?). If published, this will include your full peer review and any attached files.

Reviewer #1: **Yes: **Vinicio Barquero

Reviewer #2: No

Reviewer #3: No

---

## [Author Response · Author response to Decision Letter 0]

14 Mar 2023

Reviewer 1: We have incorporated all your suggestions into the revision. They were very helpful. 

Reviewer 2: We have incorporated your suggestions into the revision. They were very helpful and we have included detailed responses and additional information in the response to reviewers document. 

Review 3: Thank you for your suggested corrections. We have incorporated all of them into the revision.

---

## [Decision Letter · Decision Letter 1]

15 May 2023

PONE-D-22-23019R1Carbon opportunity cost increases carbon footprint advantage of grain-finished beefPLOS ONE

Dear Dr. Blaustein-Rejto,

Thank you for submitting your manuscript to PLOS ONE. After careful consideration, we feel that it has merit but does not fully meet PLOS ONE’s publication criteria as it currently stands. Therefore, we invite you to submit a revised version of the manuscript that addresses the points raised during the review process.

We look forward to receiving your revised manuscript.

Kind regards,

Malik Muhammad Akhtar, PhD, Postdoc

Academic Editor

PLOS ONE

Review Comments to the Author

Reviewer #1

Check the title.

40-43. Cite this part.

47-49. Cite this part.

89-94. These results should not be there.

130, 154, 157. Is it possible to reference the equations?

281 & 312. Please place figure title below the Fig.

320. Please include all the definitions below the table. The table must be self explanatory.

Figure S1. There are no white circles, and the map has no color

Recommendation: you may consider placing different types of dots on figure 2. This can help colorblind (daltonic) people with their understanding.

Reviewer #2

The paper conducts a pairwise comparison of pasture-finished operations and grain-finished operations. Also, the authors show that land-use intensity and carbon footprint are positively correlated. Thus, the paper concerns a fundamental and interesting research problem.

However, I have some recommendations for the authors:

1. Materials and methods.

a. Pairwise comparison: How did you collect the data? Table S8 shows information about 20 studies included in the pairwise comparison; how were the studies selected? It is necessary to add this information.

b. Regression analysis: How many observations did you use? Are the units 72 operations or 24 studies? Is it a cross-section regression, or is it panel data? (In this case, which are the individuals, and which is the time?) Where are the results of the Hausman test? Is it possible to use the Hausman test in a cross-section model? It is important to explain which model was estimated and what type of data was used.

2. Results and discussion

a. Pairwise comparison: This section describes the main findings from the t-test, but it seems that the numbers in table S3 are different from the text. Why didn’t you compare the findings from the nonparametric test with the t-test? If your data was collected from studies, why didn’t you use meta-analysis inside the t-test? I suggest explaining how the studies were selected and considering making a meta-analysis.

b. Regression analysis. Were the assumptions of the model validated? Are the same results if the authors add other variables (control)? Are the number of observations enough to estimate this kind of model?

Reviewer #3

This paper presents a rather complete comparison of the carbon footprint of grain-finished and pasture-finished beef production Systems, taking into account production emissions, soil carbon sequestration and carbon opportunity cost. The study use global information from 16 countries and several model to estimate emissions are considered. It is generally well written, with very few typing errors. The statistical methods used are correct and the authors took into account assumptions that make them valid. I recommend publishing it once a few minor corrections have been done.

Minor corrections:

Line 36. Replace “ kg” by “ kilograms”

Lines 76-76 and 80. Replace “et al” by “et al.”

Line 107 . Replace “ kg” by “ kilograms”

Line 109 111 113 Replace “et al” by “et al.”

Line 117 Repace “&” by “and”

Line 121 Replace “ kg” by “ kilograms” and eliminate “(ha)”. Universal acronyms like “ha” should not be defined

Line 125 Replace “et al” by “et al.”

Line 135 Replace “0.28 Mg carbon (C) ha-1 yr-1 “ by “0.28 Mg C ha-1 yr-1 “

Line 156 Replace “Where” by “where”

Line 162 Replace “one kg” by “one kilogram” or by “1 kg”

Lines 164 and 167 Replace “et al” by “et al.”

Line 172 Replace “ kg” by “ kilograms”

In lines 70, 76, 80 and 81 among others you write the author’s name, year, and the number of citation between brackets, but in several lines like 176, 189, 239, 259 (and more) the number in brackets is absent.

Line 176 Replace “et al” by “et al.”

Line 206 “where” without capital letter

Line 237 Replace “MgC” by “Mg C”

Line 239 Replace “et al” by “et al.”

Line 254 and 261. Replace “kg CO2e/kg” by “kg CO2e kg-1”

Line 259 Replace “et al” by “et al.”

Line 264 Table 1. Eliminate the dot at the end of table title

Line 265 Replace “ kg” by “ kilograms”

Line 320 Table 2. explain what the numbers in parentheses are

Line 353 Replace “et al” by “et al.”

Line 362 Replace “et al” by “et al.”

Line 482-483 Replace Agricultural Systems by the abbreviation Agric. Syst.

Table S4: Replace “kgCO2e kg-1” by “kg CO2e kg-1”

Reviewers' comments:

Reviewer's Responses to Questions

**Comments to the Author**

1. If the authors have adequately addressed your comments raised in a previous round of review and you feel that this manuscript is now acceptable for publication, you may indicate that here to bypass the “Comments to the Author” section, enter your conflict of interest statement in the “Confidential to Editor” section, and submit your "Accept" recommendation.

Reviewer #1: All comments have been addressed

Reviewer #2: All comments have been addressed

2. Is the manuscript technically sound, and do the data support the conclusions?

Reviewer #1: Yes

Reviewer #2: Yes

3. Has the statistical analysis been performed appropriately and rigorously? 

Reviewer #1: Yes

Reviewer #2: Yes

4. Have the authors made all data underlying the findings in their manuscript fully available?

Reviewer #1: Yes

Reviewer #2: Yes

5. Is the manuscript presented in an intelligible fashion and written in standard English?

Reviewer #1: Yes

Reviewer #2: Yes

6. Review Comments to the Author

Reviewer #1: I have reviewed the paper and I am satisfied with the changes made, please replace the figure to avoid copyright issues.

Reviewer #2: Thank you for your efforts to improve the quality of a paper that shows that land-use intensity and carbon footprint are positively correlated. The authors attended to my suggestions, so I feel this manuscript is now acceptable for publication.

7. PLOS authors have the option to publish the peer review history of their article (what does this mean?). If published, this will include your full peer review and any attached files.

Reviewer #1: **Yes: **Vinicio Barquero

Reviewer #2: No

---

## [Author Response · Author response to Decision Letter 1]

6 Jun 2023

As suggested by PLOS ONE editorial staff, we have addressed the reviewer comments regarding S1 Fig in the Supplementary Information by ensuring it follows journal guidelines. We confirmed that the basemap of the figure is in the public domain. It therefore meets PLOS ONE’s licenses and copyright policy for figures. A statement from Natural Earth providing proof that the material is in the public domain is attached as “other file” and can be viewed on their website here: https://www.naturalearthdata.com/about/terms-of-use/. 

As there were no additional reviewer comments, the manuscript, supplementary information, and figures have not been modified. Since the PLOS ONE Editorial Manager requires authors to attach a revised manuscript with and without track changes, we have attached the same manuscript and supplementary information files as we did when originally responding to reviewer comments. We apologize for any confusion this causes.

---

## [Decision Letter · Decision Letter 2]

24 Aug 2023

PONE-D-22-23019R2Carbon opportunity cost increases carbon footprint advantage of grain-finished beefPLOS ONE

Dear Dr. Daniel Blaustein-Rejto,

Thank you for submitting your manuscript to PLOS ONE. After careful consideration, we feel that it has merit but does not fully meet PLOS ONE’s publication criteria as it currently stands. Therefore, we invite you to submit a revised version of the manuscript that addresses the points raised during the review process.

We look forward to receiving your revised manuscript.

Kind regards,

Malik Muhammad Akhtar, PhD, Postdoc

Academic Editor

PLOS ONE

Journal Requirements:

Reviewers' comments:

Reviewer's Responses to Questions

**Comments to the Author**

1. If the authors have adequately addressed your comments raised in a previous round of review and you feel that this manuscript is now acceptable for publication, you may indicate that here to bypass the “Comments to the Author” section, enter your conflict of interest statement in the “Confidential to Editor” section, and submit your "Accept" recommendation.

Reviewer #4: (No Response)

Reviewer #5: All comments have been addressed

Reviewer #6: All comments have been addressed

2. Is the manuscript technically sound, and do the data support the conclusions?

Reviewer #4: Yes

Reviewer #5: Yes

Reviewer #6: Yes

3. Has the statistical analysis been performed appropriately and rigorously? 

Reviewer #4: Yes

Reviewer #5: Yes

Reviewer #6: Yes

4. Have the authors made all data underlying the findings in their manuscript fully available?

Reviewer #4: Yes

Reviewer #5: Yes

Reviewer #6: Yes

5. Is the manuscript presented in an intelligible fashion and written in standard English?

Reviewer #4: Yes

Reviewer #5: Yes

Reviewer #6: Yes

6. Review Comments to the Author

Reviewer #4: Dear Authors,

The manuscript is extremely interesting and brings a great discussion to the world stage. Some suggestions and adaptations, before final acceptance.

Reviewer #5: Acceptable as revised Acceptable as revised Acceptable as revised Acceptable as revised Acceptable as revised Acceptable as revised Acceptable as revised Acceptable as revised Acceptable as revised Acceptable as revised Acceptable as revised

Reviewer #6: I was not participating in the previous rounds of review, so I am not familiar with all discussions done so far. I apologise to the authors if some of the points below were raised before.

The paper presents and comprehensive analysis by incorporating not only production emissions but also soil carbon sequestration and carbon opportunity cost, providing a more holistic view of beef production's environmental impact.

The calculation of soil carbon sequestration (SCS) based on the mean carbon sequestration rate of 0.28 Mg C ha-1 yr-1 raises some questions. Given the considerable variation in sequestration rates due to different environmental and management factors, the paper could discuss potential implications of using a single mean rate for diverse locations. Additionally, the choice of a 100-year time frame for carbon sequestration should be justified, particularly in relation to the potential differences in sequestration rates over shorter and longer periods.

The range of total carbon footprints is wide, as indicated by the reported values ranging from -68.3 to 2169.3 kg CO2e kg-1 retail weight. While this variability highlights the diversity of beef production systems, it would be valuable to provide context and discuss potential reasons for such extreme values. Addressing potential outliers or anomalies that might significantly influence the results would enhance the credibility of the overall analysis.

While the discussion is comprehensive, some areas could benefit from further elaboration. These include the contributions of the additional components (soil carbon sequestration, carbon opportunity cost) to the overall assessment of sustainability and potential trade-offs within the components.

The paper acknowledges limitations in global representativeness due to the dataset's scope. To enhance the discussion, a more detailed exploration of how the findings might apply or differ in diverse regional contexts could be valuable. While the authors discuss limitations, further exploration of the implications of these limitations for policy and decision-making, particularly in scenarios where assumptions do not hold, could strengthen the discussion.

A question to the authors is what would happen in your analysis with countries that already have a very high carbon content in the soil (such as New Zealand due to the young soils from volcanic formation). There is limited opportunity to New Zealand to increase its soils carbon content. This is very different from countries like the USA that completed depleted soils due to crop production and are now claiming an “increase in soil carbon” due to pasture/regenerative practices. Can the USA (that depleted their soils before) get credits for soil carbon sequestration and New Zealand (that kept its soil with a high carbon content) don’t get credits? Is that a fair analysis?

Your” carbon footprint” definition does not match a Life Cycle Assessment (LCA) definition. You should be careful because readers thar don’t have an LCA background will think it does. You can’t say it is a “total” carbon footprint because soil carbon sequestration and carbon opportunity cost are included. For example, you are missing carbon sequestration from tress on beef and sheep farms – so is you carbon footprint “full”? I think it is not. You have an LCA with sensitivity analysis considering SOIL (only) carbon sequestration and carbon opportunity costs.

A better LCA term for you analysis around “land use intensity” is land occupation.

What is “retail weight beef”? Keep it simple and say it is per kg of beef. Your analysis is up to the farm-gate – you (or the authors you cite) converted the data from live weight to beef?

There is no discussion about the footprint breakdown. Grain-finished operations have a significant part of their footprint represented by carbon dioxide (a long lived GHG) due to the production of the grain. Pasture-based operation have most of its footprint represented by methane (a short-lived GHG). The authors do not address this difference in their analysis – as demonstrated by many papers published recently around the GP* metric, the emissions of methane do not corelate directly with warming if emissions are kept constant over the period of 20 years – i.e., there is no extra warming added to the atmosphere in this situation. This contrasts with the carbon dioxide that will warm the atmosphere for millennia. So this is an important difference between grain-finished and pasture finished systems that is not addressed in this paper and may affect the final results.

The authors also do not consider that, if the demand for meat stays the same, we would need to increase production in other areas -what is the full impact? Does the carbon opportunity costs are overweighed by the increase in meat production in other areas that might be less efficient (so there is a “carbon leakage”)?

7. PLOS authors have the option to publish the peer review history of their article (what does this mean?). If published, this will include your full peer review and any attached files.

Reviewer #4: No

Reviewer #5: No

Reviewer #6: No

---

## [Author Response · Author response to Decision Letter 2]

19 Oct 2023

Thank you. We greatly appreciate your careful review and suggestions. We have revised the manuscript to address the points raised. We have detailed the changes and our response to each reviewer comment in the Response to Reviewers file.

Please note that only superficial edits were made to the supporting/supplementary information: removing the term "total" from "total carbon footprint" in one table, and making the style of table titles consistent.

---

## [Decision Letter · Decision Letter 3]

15 Nov 2023

Carbon opportunity cost increases carbon footprint advantage of grain-finished beef

PONE-D-22-23019R3

Dear Dr. Daniel Blaustein-Rejto,

We’re pleased to inform you that your manuscript has been judged scientifically suitable for publication and will be formally accepted for publication once it meets all outstanding technical requirements.

Kind regards,

Malik Muhammad Akhtar, PhD, Postdoc

Academic Editor

PLOS ONE

Additional Editor Comments (optional):

Reviewers' comments:

Reviewer's Responses to Questions

**Comments to the Author**

1. If the authors have adequately addressed your comments raised in a previous round of review and you feel that this manuscript is now acceptable for publication, you may indicate that here to bypass the “Comments to the Author” section, enter your conflict of interest statement in the “Confidential to Editor” section, and submit your "Accept" recommendation.

Reviewer #4: All comments have been addressed

Reviewer #5: All comments have been addressed

2. Is the manuscript technically sound, and do the data support the conclusions?

Reviewer #4: Yes

Reviewer #5: Yes

3. Has the statistical analysis been performed appropriately and rigorously? 

Reviewer #4: Yes

Reviewer #5: Yes

4. Have the authors made all data underlying the findings in their manuscript fully available?

Reviewer #4: Yes

Reviewer #5: Yes

5. Is the manuscript presented in an intelligible fashion and written in standard English?

Reviewer #4: Yes

Reviewer #5: Yes

6. Review Comments to the Author

Reviewer #4: Dear Authors,

All corrections, questions and considerations were duly answered. I congratulate you on accepting the manuscript.

Reviewer #5: revision ok .

7. PLOS authors have the option to publish the peer review history of their article (what does this mean?). If published, this will include your full peer review and any attached files.

Reviewer #4: No

Reviewer #5: No

---

## [Editor Report · Acceptance letter]

20 Nov 2023

PONE-D-22-23019R3 

Carbon opportunity cost increases carbon footprint advantage of grain-finished beef 

Dear Dr. Blaustein-Rejto:

I'm pleased to inform you that your manuscript has been deemed suitable for publication in PLOS ONE. Congratulations! Your manuscript is now with our production department. 

Kind regards, 

on behalf of

Professor Malik Muhammad Akhtar 

Academic Editor

PLOS ONE